# USING SEMANTIC DISTANCE FOR DIVERSE AND SAMPLE EFFICIENT GENETIC PROGRAMMING

## ABSTRACT

Evolutionary methods, such as genetic programming, search a space of programs to find those with good fitness, often using mutations that manipulate the syntactic structure of programs without being aware of how they affect the *semantics*. For applications where the semantics are highly sensitive to small syntactic mutations, or where fitness evaluation is expensive, this can make learning programs intractable.

We introduce a mutation operator that yields mutated programs that are semantically far from previously evaluated programs, while still being semantically close to their parent. For function regression, this leads to an algorithm that is one to two orders of magnitude more sample efficient than other gradient-free methods, such as genetic programming, or learning the weights of a neural network using evolutionary strategies.

We show how this method can be applied to learning architecture-specific and general purpose neural network optimizers, and to reinforcement learning loss functions. The learnt components are simple, interpretable, high performance, and contain novel features not seen before such as *weight growth*.

## 1  INTRODUCTION

A program is a discrete structure, such as a symbolic expression, or sequence of instructions. Given a fitness function, genetic programming is an evolutionary algorithm for searching over the space of programs to find one with high fitness. An example of a program is a machine learning (ML) component such as an optimizer or loss function, specified using calls to a library such as TensorFlow or JAX. They contain an ordered list of nodes, where each node has an operator (such as addition or multiplication), and a list of inputs where each input is either an external input, a constant, or a previous output. These programs thus compactly describe a mathematical function that can be executed on hardware, or can be further transformed in non-trivial ways, such as being differentiated.

Can we learn such programs? An important lesson from the history of ML research is that methods that scale and leverage computation will typically eventually outperform hand-designed priors and heuristics given enough compute (Sutton, 2019). The lesson of replacing hand-designed with learnt features has often been applied to learning neural networks using gradient descent. Parts of the model such as the optimizer or loss function are important for controlling aspects such as *how fast* the model trains, but typically have less useful gradient information available to train (Metz et al., 2019) and are harder to specify using parameterized functions.

In this paper we hypothesize that one issue with learning via evolution is that mutation operators that naively manipulate the syntactic structure of a program can be highly sample inefficient, due to proposing mutations that are either too close to a previously evaluated program, or too far from the parent of the mutation (and thus likely to be deleterious). We introduce a mutation operator that uses semantic distance information to mutate safely and with diversity.

We demonstrate the effectiveness of this method in a variety of ways. For function regression, we show our method is one to two orders of magnitude more sample efficient than other gradient-free methods. We then learn architecture-specific and general-purpose optimizers, which perform well within the training distribution, and with simple, interpretable and novel features that can be used to generate research hypotheses. Finally, we learn reinforcement learning losses in grid-world environments, with novel interpretable features that transfer to Atari.

## 2 RELATED WORK

**Semantic genetic programming** See Vanneschi et al. (2014) for a survey. This includes *point mutations* and *crossover mutations*. Semantic point mutations replace a node in a program with a new one that is semantically close, but not too close (Nguyen et al., 2009). Semantic crossover mutations combine two programs by swapping two nodes which are either semantically non-equivalent, or again close, but not too close (Beadle & Johnson, 2008; Uy et al., 2009a;b). One can crossover whole programs in semantic crossover, such as summing two programs and simplifying (Moraglio et al., 2012) which can suffer from exponential size blowup; and although not genetic programming per-se, Gangwani & Peng (2017) crossover neural network policies via distillation. Our method is most similar to semantic point mutation, although crucially (1) we only use semantic information at the program level, which allows applications where per-node semantic information is not defined, and (2) we also use semantic information for *diversity* in the mutation objective.

**Diversity** Diversity in evolution usually refers to *population diversity*, where a diverse range of programs is kept in the gene pool to avoid premature convergence (Burke et al., 2004). Methods include reducing fitness for crowded areas of the phenotype space (Goldberg et al., 1987), or grouping within the phenotype space (Mouret & Clune, 2015). Our use of diversity is orthogonal to, and could be combined with population diversity: we ensure mutations are diverse from previously evaluated programs *before evaluating them*, which reduces evaluations of expensive fitness functions. Ensuring programs are not semantically *identical* to previous programs has been done before (Alet et al., 2020; Real et al., 2020) but we are the first to use semantic *distance* in the mutation objective, which generalizes to continuous search spaces.

**Learning reinforcement learning (RL) algorithms, optimizers, and loss functions** Previous work for learning loss functions and optimizers include many ways of parameterizing these components (hyperparameters, neural networks, or symbolic computation graphs), and many ways of learning (genetic programming, evolutionary strategies, meta-gradients, and Bayesian methods). Learning RL algorithms includes using meta-gradients (Oh et al., 2018; 2020) and evolving programs (Co-Reyes et al., 2021; Faust et al., 2019). Genetic programming has been used to learn loss functions (Bengio et al., 1994; Trujillo & Olague, 2006; Gonzalez & Miikkulainen, 2020). Methods for learning optimizers include meta-gradients (Hochreiter et al., 2001; Andrychowicz et al., 2016; Wichrowska et al., 2017; Lv et al., 2017; Metz et al., 2020), RL (Bello et al., 2017; Li & Malik, 2016), and evolutionary strategies (Houthooft et al., 2018). More generally, evolutionary techniques have most commonly been applied to neural architecture search (Stanley & Miikkulainen, 2002; Real et al., 2019; Liu et al., 2018; Elsken et al., 2019; Pham et al., 2018; So et al., 2019), and AutoML aims to automate the machine learning training process (Hutter et al., 2019; Real et al., 2020). Our method uses genetic programming to learn symbolic computation graphs: this has the advantage of interpretable results and avoiding noisy gradient information (Metz et al., 2019). Co-Reyes et al. (2021) is most similar to our work for RL algorithms we use semantic information in the mutation operator, but do not directly compare empirical performance due to the complexity of reproducing the same environments and evolutionary setup.

## 3 METHOD

Let $\mathcal{G}$ be the space of programs. The two key points are to define a distance function $d$ between pairs of programs which captures *semantic* information; and then given $G \in \mathcal{G}$, to define a mutation $M(G)$ which respects this distance function. This mutation can then be used in an evolutionary algorithm, for example, hill climbing where the best program so far is repeatedly mutated and evaluated.

### 3.1 THE MUTATION OBJECTIVE

Let $\mathcal{H} \subset \mathcal{G}$ be the set of previously evaluated programs, let $v(H)$ be a measure of program complexity (for example the number of nodes), and let $\mu, \beta > 0$ be constants. Our objective to minimize is

$$f(M(G)) = |d(M(G), G) - \mu| - \mu \tanh(\min_{H \in \mathcal{H}} d(M(G), H)/\mu) + \mu \beta v(M(G)). \tag{1}$$

This objective contains three terms.

1. The first term requires that $d(M(G), G) \approx \mu$ for some *mutation rate* $\mu > 0$; i.e., that $G$ and $M(G)$ are semantically similar (which means that the mutation is less likely to be deleterious), but not too similar (since evaluating almost identical programs would also lead to wasted computation). This is similar to similarity-based mutation (Uy et al., 2009a;b), although we only ever consider distances between programs rather than nodes within a program: this allows us to work with programs where internal nodes do not have meaningful semantics by themselves, such as in optimizers or loss functions.

2. The second term encourages *diversity* by penalizing programs that are close to a previously evaluated program. Note that the function $d \mapsto \mu \tanh(d/\mu)$ is approximately the identity when $|d| < \mu$, and is otherwise bounded from above by $\mu$, thus this term gives a "repulsive force" between $M(G)$ and $H$ only for $H \in \mathcal{H}$ where $d(M(G), H) \lesssim \mu$. We use $\min$ rather than $sum$ since we want the first two terms to be balanced.

3. The third term penalizes programs that are complex, and is similar to parsimony pressure (Koza, 1992; Zhang & Mühlenbein, 1995); this is to counteract the phenomena of *bloat* where the number of nodes grows without a corresponding increase in fitness.

**Distance function**   What distance function $d$ should we use? We build the distance function by mapping each program to a vector, and then using the $L_1$ distance (i.e., mean absolute difference) between vectors. More precisely, we build $\mathbf{s} : \mathcal{G} \to \mathbf{R}^m$ (which we refer to as the **semantics** of a program), a scaling $p(G) \in \mathbf{R}^m$ where $G$ is the graph to mutate, and define the distance between $J, K \in \mathcal{G}$ to be

$$d(J, K) = \frac{1}{m} \sum_{i=1}^{m} p_i(\mathbf{s}(G)) \cdot |\mathbf{s}(J)_i - \mathbf{s}(K)_i|. \tag{2}$$

For all our applications, the semantics relies on program input samples $(x_i)_{i=1}^n$ where each $x_i$ is some input to a previously evaluated program (see §3.3). Note that although the distance function $d$ is a function of $G$ and the $x_i$ and so varies as evolution proceeds, it in practice tends to vary slowly.

The best scaling and semantics mapping depend on the application. For example, in function regression in §4 where the program $H$ is used as a scalar-valued function $H : \mathbf{R}^k \to \mathbf{R}$, we use the *identity semantics* $\mathbf{s}_{\text{id}}(H) = (H(x_1), \ldots, H(x_n)) \in \mathbf{R}^n$ and the scaling $p = 1$. If $H$ is used as a loss function, the *gradient semantics* $\mathbf{s}_{\text{grad}}(H) = (\nabla_x H(x_1), \ldots, \nabla_x H(x_n))$ is more appropriate, since this does not distinguish between programs whose outputs differ by a constant, but does take into account stop-gradient nodes.

## 3.2   IMPLEMENTATION DETAILS: MUTATING COMPUTATION GRAPHS

In our applications we restrict to programs in the form of computation graphs, i.e., ordered lists of nodes, where each node has inputs from previous outputs or external inputs, and performs some operation such as addition. We minimize (1) by generating several candidate mutations, and then use CMA-ES to fine-tune the constants.

**Generating raw mutations**   See Appendix A for details. In the first step, generate several candidate programs $G_1, \ldots, G_k$ (we take $k = 16$) by repeatedly taking the original program $G$, then applying up to two changes to it:

1. With probability $0.5$ adjust a uniformly chosen node. Adjusting a node $z$ involves replacing it with one of the following expressions sampled uniformly: $\{zc, z + c, z + cf, z(1 + cf), z/(1 + c|f|)\}$, where: $c \in \{0, 1\}$ is a "gating" constant that allows continuous linear interpolation between the old and new node and is initially set to a value such that the new node has the same value as the old node; and $f$ is either an existing node, or a newly constructed node which is a randomly chosen function of existing nodes. Thus after this step, the program still has the same semantics as the previous program.

2. With probability $0.5$, replace a randomly chosen node with one of its inputs. This allows program simplification.

We simplify the resulting program, for example by removing unused nodes, and merging adjacent add and multiply nodes. We repeat these steps until $k$ distinct programs (ignoring constants) have

been generated. We experimented with different raw mutation operators, mutation probabilities, and the number of mutations applied, but it did not make a noticeable difference to performance.

**Optimizing** The second step uses the black-box optimization algorithm CMA-ES (Hansen & Ostermeier, 2001; Lange, 2022) to optimize the constants appearing in each of the programs $G_i$ to minimize (1); the program with the smallest value of (1) is then the mutation output. We use a CMA-ES population size of 128, initial standard deviation of 0.01, and run for 200 generations. (Thus we evaluate (1) a total of $16 \times 128 \times 200 \approx 10^6$ times, which is fast on modern ML hardware; this is one source of how we are able to achieve orders of magnitude speed-up; see §4.2.)

To make optimization more stable, CMA-ES does not directly tune the constants (which can have dramatically different magnitudes). Instead, for a candidate program, let $u \in \mathbf{R}^t$ be the initial constants appearing in the computation graph, let $v \in \mathbf{R}^t$ be the values that CMA-ES is tuning; we set the new program constants to be $u + r \odot v$ where $r \in \mathbf{R}^t$ is custom scaling, defined as $r_i = \max\{10^{-5}, \min\{|u_i|, |u_i - 1|, |u_i + 1|\}\}$. Observe that when the initial value $u_i$ of a constant is large, then CMA-ES "step size" will be much bigger than if it is close to 0, or a multiplicative constant close to 1 such as 0.999.

### 3.3 Implementation details: distributed setup

We evaluate programs in a distributed setup consisting of a single central controller, and several evaluation workers (ranging from 8 to 64 in our experiments) each consisting of a machine with TPUv2-8 Tensor Processing Units.

**Sampling and/or mutation** The central controller keeps track of the fitness of evaluated programs. When a evaluation worker is free, it requests a program to evaluate. A program is selected from those with highest estimated fitness (or the original if none have yet been evaluated), mutated on the evaluation worker, and evaluated. (Additionally for applications with noisy fitness measurements, such as RL in Section 7, with probability 0.2 we re-evaluate the program without mutating it; this reduces fitness uncertainty.)

We found it beneficial to bias the selection of the program towards simpler programs, by first sampling the max number of nodes uniformly from the node count range already seen, and then sampling the best program conditioned on having at most this many nodes. This can be seen as maintaining diversity in complexity, and as such is similar to Mouret & Clune (2015).

**Mutation rates and parsimony pressure** We resample $\mu$ and $\beta$ for (1) in each mutation. The optimal value of $\mu$ depends on the fitness landscape around $G$; for example, if we are close to optimum, then smaller $\mu$ will likely work better. $\beta$ trades off between program complexity and the distance terms, and there is likely no single value that works better either. Rather than picking a single fixed value, we sample $\mu \sim U(0, 0.4)$ and $\beta \sim \exp(U(-6, -1))$ where $U(a, b)$ is a uniform random variable in the range $(a, b)$. These values were chosen using hyperparameter sweeps on the function regression and optimizer (with a reduced set of tasks) applications.

**Collecting sample inputs for the semantics** Recall that the semantics function **s** is defined in terms of sample program inputs $(x_i)_{i=1}^n$. These are collected by the evaluation workers, and stored in the central worker, which uniformly samples the $x_i$ from across all evaluated programs.

## 4 Analysis: learning simple functions

For initial analysis we learn a program $G$ that approximates some unknown function $f$, where the fitness is one minus the mean square error over 1000 uniformly sampled points $y_i$, i.e.

$$F_f(G) = 1 - (\mathbf{E}_i y_i (G(y_i) - f(y_i))^2)^{1/2}. \tag{3}$$

We evaluate on $f(y) = \sin(y)$ for $y$ in the range $[-\pi, \pi]$ and $[-2\pi, 2\pi]$, and $f(y, z) = \text{atan2}(y, z)$ for $y, z$ in the range $[-1, 1]^2$, since these cover a range of function complexities, with both scalar and vector inputs. For the distance function (2) we use the identity semantics and $p = 1$. We restrict graph operators to add, sub, mul, div, abs, and max.

### 4.1 PERFORMANCE AND ABLATIONS

Figure 1 shows the performance according to which terms of the mutation objective (1) are included. Diversity is very important for fitness, and simplicity is mildly important. One hypothesis for why simplicity also helps fitness is that since graphs with fewer nodes give a lower dimensional set of functions, there are simply fewer *distinct* graphs to explore over. Figure 2a shows the average number of nodes in the graph with the highest fitness over time; interestingly both the diversity *and* simplicity terms are required for obtaining small graphs. Figure 2b shows the effect of the diversity term: the new mutation is pushed to be distinct from nearby previously evaluated functions.

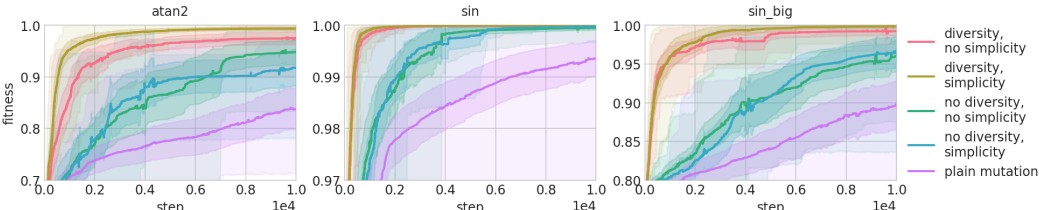

Figure 1: Performance with and without the diversity and/or simplicity terms of (1). Also shown is "plain mutation" which is the raw mutation operator of §3.2 without optimizing for any objective.

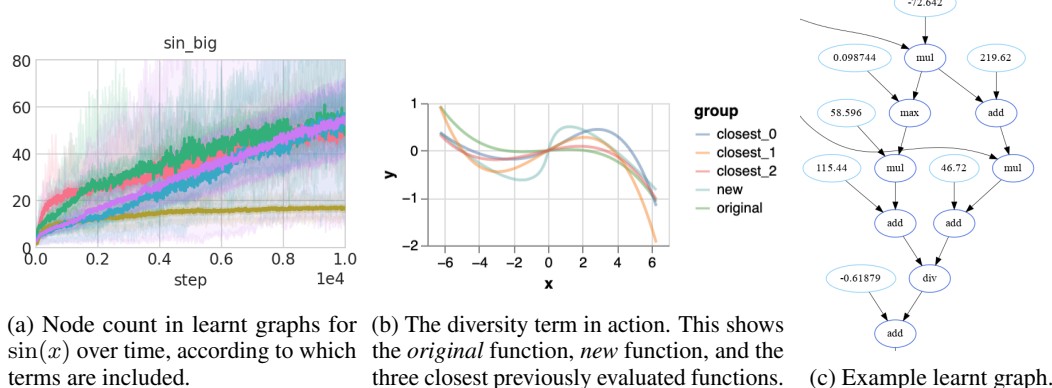

(a) Node count in learnt graphs for $\sin(x)$ over time, according to which terms are included.

(b) The diversity term in action. This shows the *original* function, *new* function, and the three closest previously evaluated functions.

(c) Example learnt graph.

Figure 2: Insights into the algorithm.

### 4.2 SAMPLE EFFICIENCY COMPARISON

We compare our method with other approaches for learning a function given scalar fitness measurements, including plain point mutation with and without crossover, other semantic based methods, and also using CMA-ES to learn a parameterized function. See Appendix B for details. We also tried gplearn (Stephens, 2016), which learns tree-based programs, but it had much worse performance. As Figure 3 shows, we are one to two orders of magnitudes more efficient than other methods.

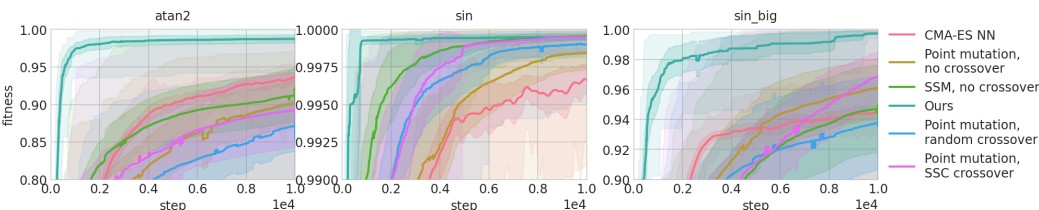

Figure 3: Comparison in fitness sample efficiency. CMA-ES NN learns the weights of a neural network through evolutionary strategies. Point mutation is the raw mutation algorithm described in §3.2 with parsimony pressure. SSM is Semantic Similarity Mutation (Uy et al., 2009a). Random crossover is randomly swapping nodes. SSC is Semantic Similarity Crossover (Nguyen et al., 2009).

# 5 EXTENDING TO SEMANTIC CROSSOVER

We briefly remark we can generalize this approach to create a crossover operator $C$, and we show how. Given given two programs $G, H \in \mathcal{G}$, write $C = C(G, H)$, and let

$$f_{\text{crossover}}(C) = \max\{d(C, G), d(C, H)\}. \tag{4}$$

This objective is minimized when $C$ is semantically halfway between $G$ and $H$. This can be minimized following the same approach as in §3.2. This further improves performance; see Figure 4.

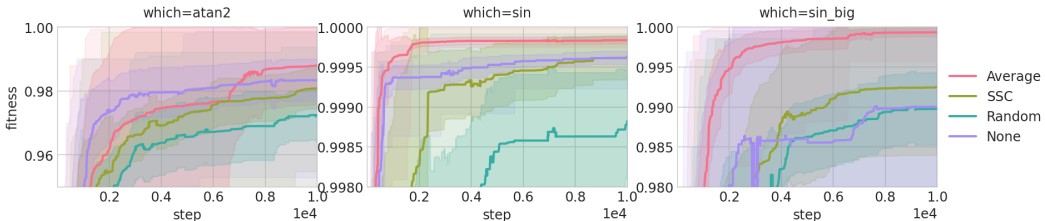

Figure 4: This shows a sample efficiency when the mutation operator described by (1) is used in conjunction with crossover. *Average* is the semantic crossover operator described by (4). *SSC* is swapping semantically similar nodes (Uy et al., 2009a). *Random* swaps nodes at random.

# 6 APPLICATION: LEARNING AN OPTIMIZER

We demonstrate the method on non-toy tasks. Can we learn a neural network optimizer? An elementwise *optimizer* is a function $f : \mathbf{R} \times \mathbf{R} \times \mathbf{R}^k \mapsto \mathbf{R} \times \mathbf{R}^k$ that maps $f : (g, \theta, s) \mapsto (\theta', s')$, where $g \in \mathbf{R}$ is a gradient, $\theta, \theta' \in \mathbf{R}$ are the previous and new parameter values, and $s, s' \in \mathbf{R}^k$ are the previous and new state values for some $k \geq 0$. For example, gradient descent with momentum has $k = 1$ and $f(g, \theta, m) = (\theta - \eta m', m')$ where $m' = (1 - \kappa)m + \kappa g$ for some $0 < \kappa < 1$. By transforming $f$ by an additive constant, it is always possible to take the initial state to be $0 \in \mathbf{R}^k$, and so this formalism captures all optimizers that act elementwise, such as Adam (Kingma & Ba, 2014).

Recall that we need to define **optimizer semantics** to define distance between programs. The identity semantics applied to $f$ is inappropriate, since we care about the (long term) effect on parameters rather than the immediate change in optimizer state. So we instead build the semantics as follows:

1. The samples $(x_i)_{i=1}^n$ (see §3.3) used to construct the semantics are pairs $x_i = (\theta_i, (g_{ij})_{j=1}^s)$, where $\theta_i \in \mathbf{R}$ is the value of some parameter at some point during training and $(g_{ij})_{j=1}^s$ is a sequence of gradients for that parameter at subsequent timesteps. We collect $n = 500$ trajectories each of length $s = 2000$.

2. For each $x_i$, let $\Theta(G, x_i) \in \mathbf{R}^{s+1}$ be the parameter rollout starting at $\theta_i$ and applying optimizer updates to it using the gradients $g_{ij}$ (we set the initial optimizer state to 0).

3. Define the semantics to be the set of single step parameter changes at 1000 randomly sampled points. I.e., sample $1 \leq i_1, \ldots, i_{1000} \leq n$ and $1 \leq j_1, \ldots, j_{1000} \leq s$ and set

$$\mathbf{s}_{\text{opt}}(G) = (\Theta(G, x_{i_k})_{j_k} - \Theta(G, x_{i_k})_{j_k-1})_{k=1}^{1000} \in \mathbf{R}^{1000}. \tag{5}$$

The semantics in (5) is a sequence of parameter value changes. These may be dramatically different in magnitude, so for the scaling $p$ in (2) we normalize these by

$$p(x) = 2/(|x| + \text{median}(|x|)) \tag{6}$$

where $|x|$ is the elementwise absolute value of $x$. For mutation, in addition to the operator described in §3.2, we also allow changing the state size $k$, although in the experiments below our best performing optimizers did not make use of this.

## 6.1 LEARNING A GENERAL PURPOSE OPTIMIZER

Can we learn a hyperparameter-free optimizer that achieves good eval performance across a wide range of network architectures and datasets? We measure fitness by training and evaluating on the

set of fixed tasks $\mathcal{T}$ in the learnt optimization task suite[1], with the hope that an optimizer trained on sufficiently many tasks will generalize. Each task $T \in \mathcal{T}$ defines a parameter initialization function, train and eval datasets, and a loss function. The fitness of an optimizer $G$ is then given by (7), where eval_loss$(G, T)$ is the evaluation loss after training for $10^4$ steps, $l_T < L_T$ are constants hand-picked by examining the typical loss range, and train_frac_finite$(G, T) \in [0, 1]$ is the fraction of train steps that the train loss is finite, which gives a signal on tasks which diverge during train.

$$F_{\text{opt}}(G) = \frac{1}{|\mathcal{T}|} \sum_{T \in \mathcal{T}} \Big( \max \big(0, \frac{L_T - \text{eval\_loss}(G, T)}{L_T - l_T}\big) + \text{train\_frac\_finite}(G, T) - 1 \Big) \quad (7)$$

We set the initial optimizer equal to momentum with a learning rate of $10^{-3}$. We use 64 parallel evaluation workers, with each fitness evaluation taking approximately 30 minutes.

**Results** The initial optimizer has a fitness of $0.331$. After approximately 3500 optimizer evaluations, the following optimizer with a fitness of $0.551$ was discovered, with a single state variable $m$ (initialized at zero as described above) and parameter $\theta$ update formula

$$m' = \frac{g + m}{1.0002 + 10.881|g|} \quad (8)$$

$$\theta = 0.99986(\theta - 0.01636m') + 2.67 \times 10^{-6} \quad (9)$$

This optimizer performed well across all tasks, with average performance approximately equal to that of Adam (which gets a fitness of $0.554$), but with only a single state variable and far fewer nodes. The $0.99986$ term is L2 parameter decay. However the "momentum" update (8) has very interesting properties, and to the best of our knowledge has not been seen before. Rather than scaling momentum uniformly at every step, it only scales down by large gradients; in particular parameters that receive small gradients accumulate "momentum" over more steps. Additionally the change in $\theta$ is always bounded, since $|m| \leq 1/10.881$ no matter how large $|g|$ is.

Interestingly, this optimizer failed to generalize outside the task suite, failing to train a large transformer. This was unexpected, since the optimizer has fewer constants than the number of tasks $|\mathcal{T}| \approx 50$, so we did not expect it to overfit, but it is possible that the tasks in $\mathcal{T}$ lack diversity, for example the length of training, or gradient magnitude. Nonetheless this is a promising initial result: (1) we are able to learn a simple symbolic optimizer that performs well within training distribution, so we are reduced to the problem of finding a representative set of tasks $\mathcal{T}$, (2) the optimizer contains novel features not seen before, which can generate interesting research insights in themselves, and (3) there are many potential avenues to be explored, such as allowing the learnt optimizer to make use of extra inputs such as gradients from other parts of the network.

## 6.2 LEARNING A NETWORK-SPECIFIC OPTIMIZER

We also learn an optimizer to fine-tune a single model: CIFAR-10 classification using VGG (Simonyan & Zisserman, 2014). The fitness is the evaluation accuracy after $8 \times 10^4$ training steps. To allow for discovery of things like learning rate decay, we provide a scalar input $t \in [0, 1]$, which is the fraction through training. We set the initial optimizer equal to momentum, and the best performing optimizer is discovered after around 1000 evaluations:

$$\theta' = (0.9992 + 0.001t)(\theta - 0.0012g) \quad (10)$$

This is a very simple optimizer (momentum has been unlearnt). But it is high performing: it matches the performance of a highly tuned momentum plus learning rate-schedule with weight decay. The term $(0.9992 + 0.001t)$ corresponds to an L2 parameter regularization term, i.e., weight decay. However, it varies with time. At $t = 0$ it corresponds to a loss term of $8 \times 10^{-4}\theta^2/2$; and more interestingly at $t = 1$ it corresponds to a loss term of $-2 \times 10^{-4}\theta^2/2$, i.e., the final 20% of training the network actually undergoes *weight expansion*. This is not a superfluous feature, as shown in Figure 5 weight expansion helps final performance, and is an example of the sorts of research insights that can be generated using evolved computation graphs.

---

[1]http://github.com/google/learned_optimization

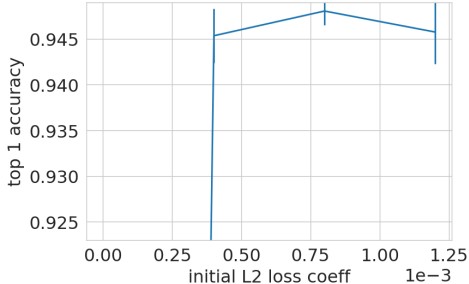 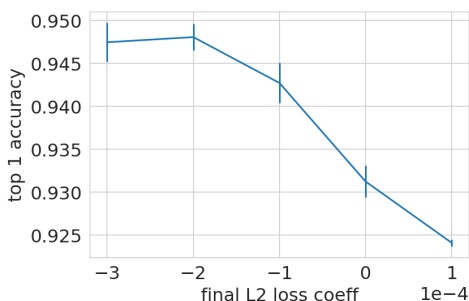

Figure 5: Final eval accuracy according to initial and final L2 parameter coefficients. Having a positive L2 loss coefficient is required through most of training; however, a slight negative L2 loss coefficient (weight expansion) at the end of training helps performance; this is not something we are aware has been investigated before. (Error bars show 1 standard deviation, over 3 seeds.)

## 7  APPLICATION: LEARNING A REINFORCEMENT LEARNING LOSS

A typical deep reinforcement learning agent contains a neural network that maps an observation to a policy and a value. The parameters of the network are trained using a loss function, which takes in these predictions together with other inputs such as the environment rewards. The loss function affects the agent performance, with innovations such as reward scaling, handling off-policy data Espeholt et al. (2018), trust region policy updates (Schulman et al., 2015), etc leading to better performing agents. Can our method can learn loss functions that are effective across a range of environments?

**Loss graph and semantics**  We use the following notation. At time step $t$, the agent network outputs policy $\pi_t$ and value $V_t$ given an observation $s_t$. An action $a_t$ is sampled from $\pi_t$, and the agent then receives a reward $r_t$, with the mask $m_t \in \{0, 1\}$ indicating that the environment has not yet terminated. We restrict to loss functions that can be expressed using a backward trajectory with $n$-step bootstrapped returns $R_t$. The graph $G$ defines a per-step loss in the $n$-step trajectory, where $n = 20$. We additionally input the policy entropy $H_t = -\mathbf{E}_{a \sim \pi_t} \log \pi_t$, since reduction nodes in the computation graph are not yet supported.

$$G : (V_t, V_{t+1}, r_{t+1}, m_t, m_{t+1}, \log \pi_t(a_t|s_t), H_t, R_{t+1}) \mapsto (\text{loss}, R_t) \tag{11}$$

For the semantics we collect 500 input trajectories (i.e., the left hand side of (11)). Given a graph $G$, for each trajectory calculate the gradient of the loss defined by $G$ with respect to the three differentiable inputs; this yields three vectors $\frac{\partial L}{\partial V}, \frac{\partial L}{\partial H}, \frac{\partial L}{\partial \pi} \in \mathbf{R}^{500 \times 20}$. We then randomly sample 1000 points from each of these vectors, yielding $\mathbf{s}_{\text{RL}}(G) \in \mathbf{R}^{3 \times 1000}$. We use the scaling $p$ defined by (6).

**Evaluation**  We evaluate the learnt loss function on the 15 environments introduced by Oh et al. (2020), which consist of 5 tabular grid worlds, 5 random grid worlds, and 5 delayed chain MDPs. These environments are small and implemented in JAX (Bradbury et al., 2018) which allows quickly training agents with the environment in the loop. We use approximately the same network architecture, environment settings, and number of steps per lifetime as in (Oh et al., 2020), except we cap at a maximum of $10^7$ steps per task for speed of training. The fitness is defined as the average scaled episode return across the different environments after training.

As a baseline, we use advantage actor critic (A2C) (Mnih et al., 2016) with a discount factor $\gamma$ and loss coefficient $c_{\text{entropy}}$. This is defined by the following equations:

$$R_t = r_{t+1} + \gamma m_{t+1} \text{sg}(V_{t+1}) \tag{12}$$
$$A_t = R_t - \text{sg}(V_t) \tag{13}$$
$$\text{loss} = -A_t \log \pi(a_t|s_t) + \frac{1}{2}(R_t - V_t)^2 - c_{\text{entropy}} H_t, \tag{14}$$

where $A$ is an advantage, and sg denotes a stop-gradient operator. Our graph was initialized to the baseline A2C and evolved from it.

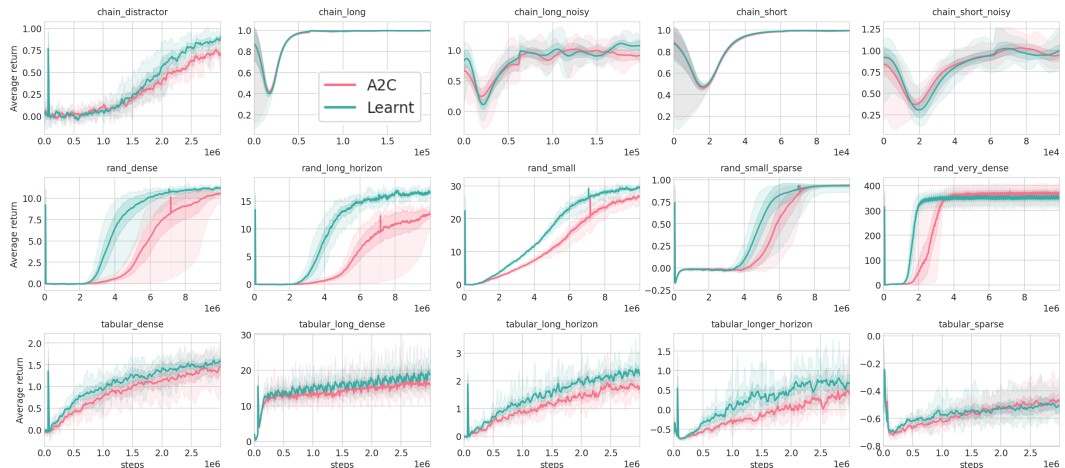

Figure 6: Performance of the best performing learnt loss function and a tuned A2C baseline on 15 training environments from (Oh et al., 2020).

**Results**    Figure 6 shows our loss function outperformed the baseline A2C across 15 environments.[2]

The learnt loss function made various changes to the original A2C loss function, including discovering TD($\lambda$) returns (Sutton & Barto, 2018). It also made the following simple change to the policy gradient term, namely replacing the advantage coefficient in Eq (14) with

$$A' = A + 3.15 \cdot \max(0, A).$$

This has the effect of more heavily weighting policy updates to when the advantage is positive rather than negative (i.e., actions with higher-than-expected returns). In fact, this term was proposed by Oh et al. (2018) as self-imitation learning combined with A2C. We confirm the learnt loss function transfers to Atari: a modified version of it (replacing the 3.51 constant with 1) on Impala (Espeholt et al., 2018) trained for 200M frames without any further hyperparameter tuning performed better than A2C on 26 out of 57 games and worse on 22 out of 57 games. On 5 games it significantly improved the performance of A2C (see Appendix C). These results shows that our method is capable of both performing well within-training distribution, as well as generating research insights due to the interpretable nature of the learnt programs.

## 8    CONCLUSION

We have shown how to use whole-program semantic information in a mutation objective to achieve safe, diverse mutations. This method significantly improves sample efficiency, allowing us to discover algorithmic features for optimization and reinforcement learning. These methods may generalize to other program spaces too, for example neural architecture search.

There are some limitations of the method. The use of a distance function requires engineering infrastructure to collect program inputs. Although not specific to our approach, evolution in general is still data intensive, and the solutions discovered are limited to the search space, but we hope that as the amount of compute and how the programs are used increases, that these become less of an issue, in the same way that neural networks are sufficiently flexible to capture many transformations we care about.

The experiments we have ran so far have been relatively limited, with small amounts of data (the largest experiments trained for less than two days), on small tasks such as neural networks with only a few layers and grid-world RL environments. The methods can be easily scaled up with more compute, across more tasks, and for longer, as well increasing the expressivity of the program (such as reduction operators) and how the programs are used, and we hope that this will lead to increasingly powerful programs being discovered.

---

[2]Although we use approximately the same set of environments as Oh et al. (2020), the settings were slightly different. Thus we compare against our own tuned A2C baseline to which our program was initialized.

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

## A GENERATING RAW MUTATIONS

We found that the precise form of the raw mutation operator which generates candidate programs to fine-tune did not matter, i.e., that the performance of the final algorithm was largely insensitive to precise implementation details and probabilities. But for completeness we detail the raw mutation operator used in our experiments.

Given an the original program $G$, we generate a mutation by the following procedure.

1. With probability $0.5$, do the following. Select a node $z$ uniformly at random. Select an expression from $\{zc, z + c, z + cf, z(1 + cf), z/(1 + c|f|)\}$ uniformly at random. Here, $c$ is a constant, that is chosen from $\{0, 1\}$ to be such that the resulting expression is equal to $z$ ($c$ is later fine-tuned to optimize the mutation objective.) Further, $f$ is selected as follows. With probability $0.5$, $f$ is an existing node in the program selected uniformly at random. And otherwise, $f$ is generated by first uniformly selecting a function in {abs, add, div, exp, log, max, min, mul, neg, pow, sqrt, square, sub, cos, cosh, sin, sinh, tan, tanh}, and then selecting the appropriate number of inputs for the arity of the function uniformly from the existing nodes in the program.
2. With probability $0.5$, replace a randomly chosen node with one of its inputs. This allows program simplification.

We simplify the resulting program, for example by removing unused nodes, and merging adjacent add and multiply nodes.

## B DETAILS OF FUNCTION REGRESSION COMPARISONS

We describe here the methods compared in Figure 3.

### B.1 SEMANTIC SIMILARITY CROSSOVER (SSC)

This method follows Nguyen et al. (2009). For two programs represented as computation graphs, we calculate the semantic distance between each pair of nodes (one in each program) by using the collected sample program inputs $(x_i)_{i=1}^n$ and taking the L2 distance between the intermediate outputs. We then uniformly sample two nodes to crossover among nodes whose distance is within the range $[0.1, 1.0]$ (chosen using a hyperparameter sweep), or uniformly sample between all pairs if there are no nodes within this range.

### B.2 SEMANTIC SIMILARITY MUTATION (SSM)

This method follows Uy et al. (2009a). Given a program represented as a computation graph, we first select a node uniformly at random, and then generate candidate mutations using the same method as described in 3.2 to find one which is at an L2 distance of $\mu$ from the original node, and then replacing the original node with the new (sub) computation graph. Here, $\mu \sim U(0, 0.4)$ as in Section 3.3.

### B.3 LEARNING THE WEIGHTS OF A NETWORK USING CMA-ES

We use CMA-ES (Hansen & Ostermeier, 2001; Lange, 2022) to tune the weights of a fully connected neural network. The following hyperparameters, obtained using a sweep, were used:

- Number of intermediate hidden layers = 2
- Total parameter count = 128
- CMA-ES Population size = 16
- CMA-ES Initial standard deviation = 0.01

## C GRAPH VISUALIZATION: A2C, AND LEARNT GRAPHS

## D POSITIVE-ADVANTAGE-BIAS ON ATARI

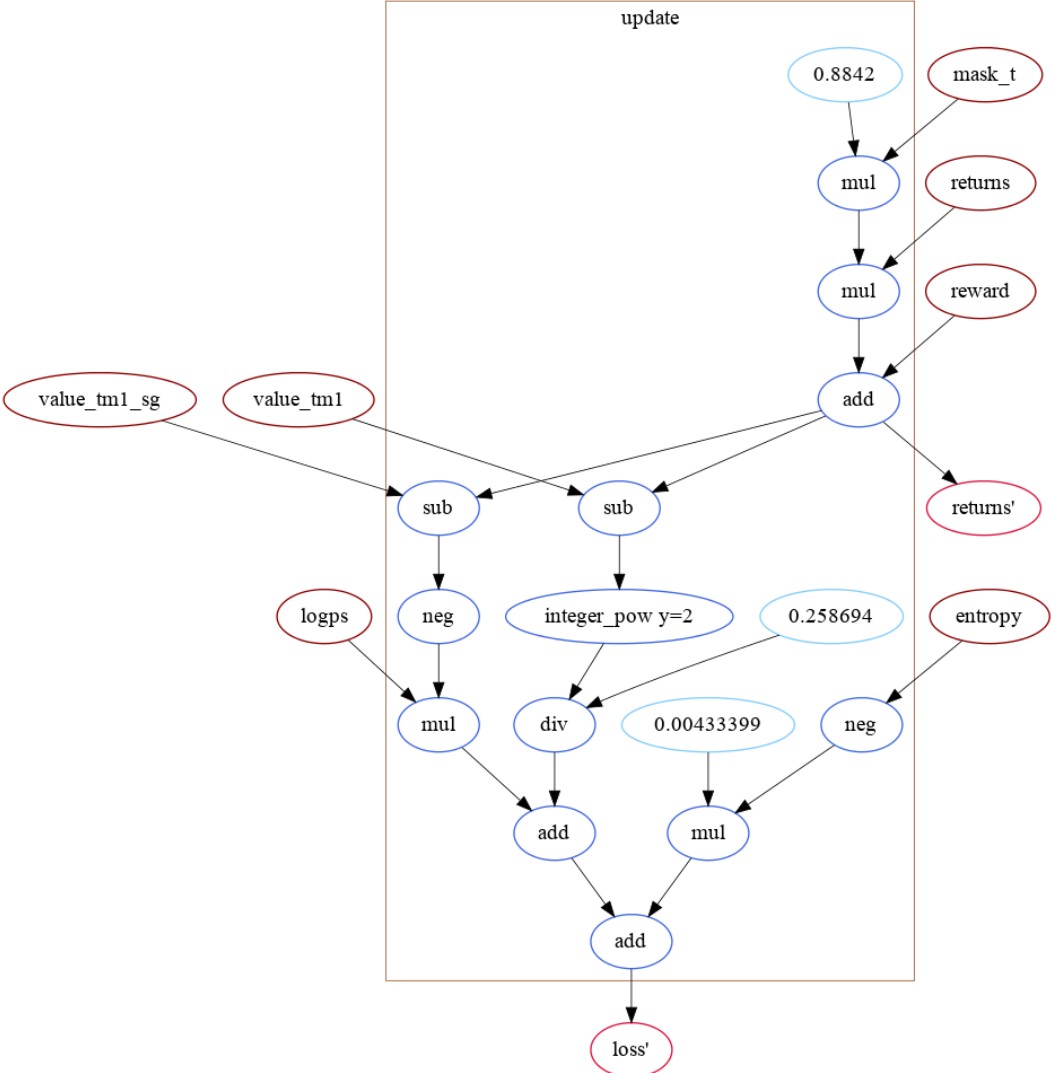

Figure 7: The A2C graph. This shows one with constants tuned for maximizing the RL fitness (but no nodes added or removed). The "_sg" suffix indicates a stop gradient has been applied. Performance is shown in Figure 6 as *A2C*.

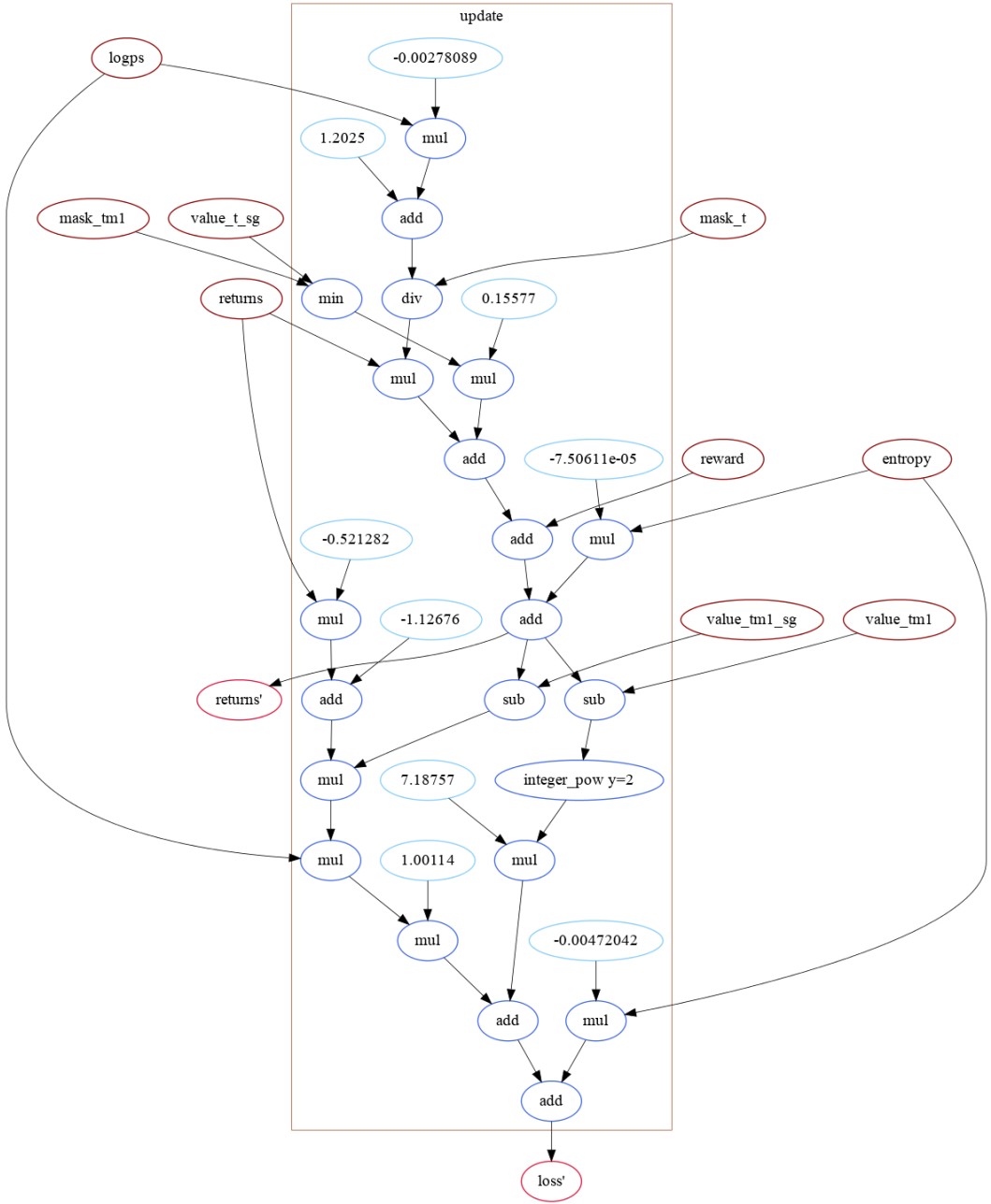

Figure 8: A graph learnt from a search initialized with A2C (see Figure 7). The "_sg" suffix indicates a stop gradient has been applied.

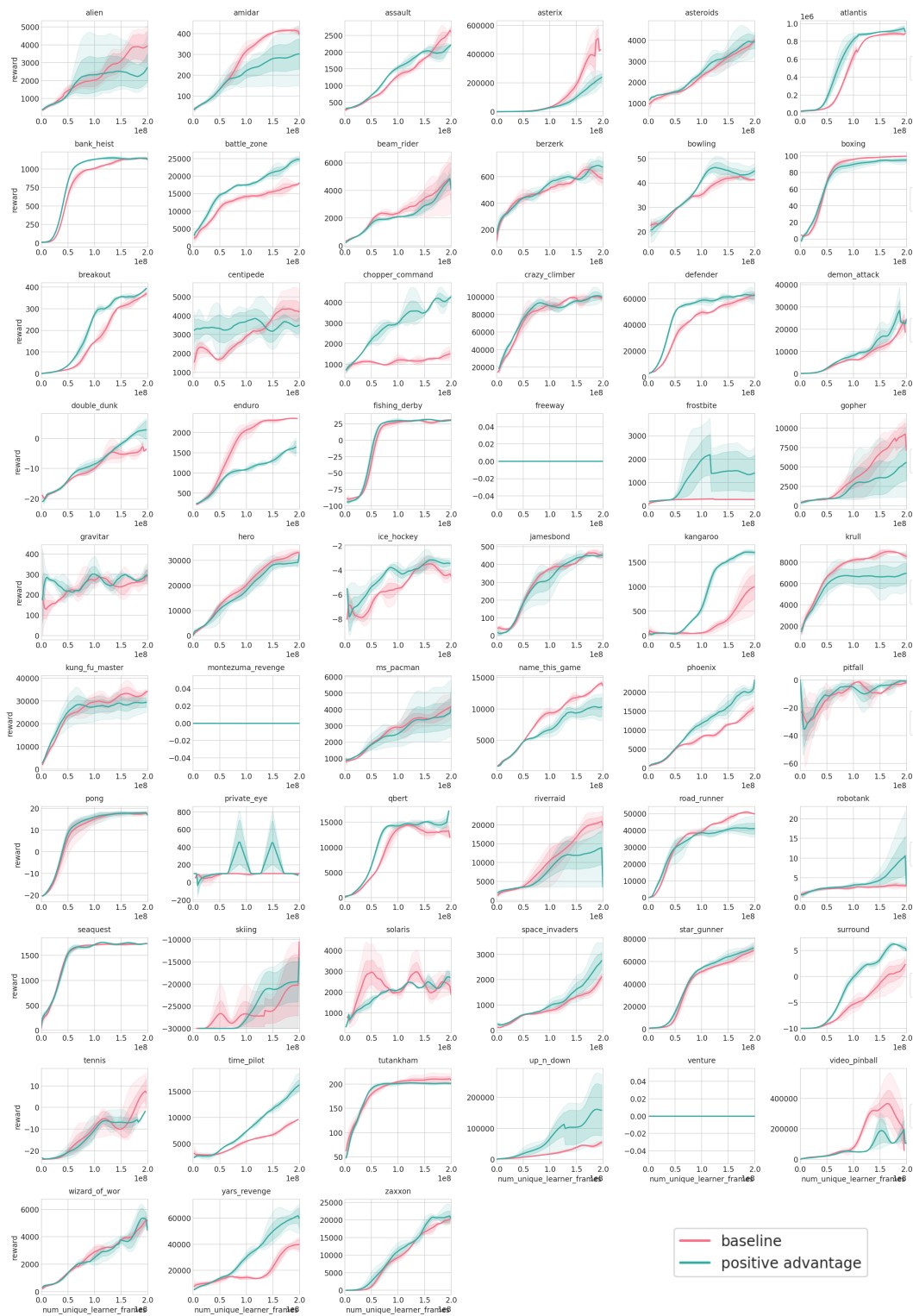

Figure 9: Performance with the advantage term used in the policy gradient loss in Impala (Espeholt et al., 2018) replaced by advantage + max(0, advantage). Trained over 200M frames, 3 seeds each.

