# OpenReview forum: "Using semantic distance for diverse and sample efficient genetic programming"
_ICLR.cc/2023/Conference — Submitted to ICLR 2023_

### Official Review · Reviewer_zufj · 2022-10-27

**Confidence:** 4
**Correctness:** 2
**Technical Novelty And Significance:** 2
**Empirical Novelty And Significance:** 2
**Recommendation:** 1

**Clarity, Quality, Novelty And Reproducibility:**

- Sec. 3.1, presenting the mutation objective is relatively clear, although quite abstract, it presents the approach in terms of a generic distance measure.
- The generic function is detailed in Eq. 2 as a scaled L1 norm over the semantic variables, but still an abstract scaling function p_i(s(G)), where G is not clearly defined given that the distance is stated as d(J,K), so between J and K. What’s G, and what’s its use? According to Eq. 1, the G is in fact used as J=M(G) and K=G for Eq. 2. I’m confused by this notation, better explanations are required.
- The second term of Eq. 1, which is the diversity term, appears quite ad hoc. I get it does some kind of adjustment for getting diverse solutions, but is there any more formal or conceptual justifications for that part? I mean, why not just summing the L2 norm with all individuals in the current population as diversity?
- Is the scaling between the three components of Eq. 1 with \mu parameter enough and allow the best combination of these components. The scale may vary between these, would a distinct scaling factor between them allow better performances?
- It is stated at the beginning of Sec. 3.2 that “We minimize (1) by generating several candidate mutations, and then use CMA-ES to fine-tune the constants.”. Ok, but how many individuals are we generating before picking the best one according to Eq. 1? And what is the effect of the number of individuals generated on the quality of the mutations?
- In Sec. 3.2, details are missing on the raw mutation. For instance, how the choice of the replacement of z with the expressions done uniformly? How is the new node used and constructed? Is the new node replacing the current node directly, or a new subtree is made? And how about the arity, do we restrict the new node to have the same arity than the old one? If not, how do we proceed to enforce validity of the tree? Many details on that mutation operator are missing.
- Also, still in Sec. 3.2 on the raw mutation, it is stated that “c\in{0 1} is a “gating” constant that allows continuous linear interpolation between the old and new node and is initially set to a value such that the new node has the same value as the old node.” I see that this is the case for c=0, but how can this be enforced for c=1? (Conversely for zc expression, where the semantic is preserved for c=1, but not for c=0).
- In the “Optimizing” paragraph, it is stated that s_i = max{10^-5,min(|u_i|, |u_i-1|, |u_i+1|)}. What do the indices i stand for? Why set the u in +- 1 range? I don’t get this part, the tricks of that equation are unclear to me, I am not sure how this helps to handle large or small constants.
- In Sec. 3.3, paragraph “Sampling and/or mutation”, it is stated “With probability 0.2, this is re-evaluation, which allows reducing the fitness measurement error for potentially high fitness programs”. Wait, what is that? I mean, if you have the fitness of an individual that is not modified, you don’t need to re-evaluate it. If it is modified, then you should re-evaluate. I don’t get why and for what purpose we would re-evaluate some individuals picked at random, some explanations are required here.
- In the “Mutation rates and parsimony pressure”, it is stated that \mu and \beta are picked randomly according to some distribution for each mutation. That’s quite fancy, this is an extra layer of complexity over the proposed mutation operator. I would like to see an ablation study over these to demonstrate the added value of proceeding that way.
- Paragraph “Collecting sample inputs for the semantics”, it is stated that “the evaluation workers also collect program inputs when evaluating a program”. Why doing so, what’s the use? How does this affect the optimization process?
- Performance and ablation in Sec. 4.1 is interesting to show the effect of diversity and simplicity, but these results are for one case. I would have been good to conduct this ablation over several problems.

Using CMA-ES for constant optimization is a big deal in my opinion, as (200x128)=25600 fitness evaluations are spent on making this tuning for each mutated individual. We usually count the sample efficiency of GP in term fitness evaluation, and as such, this is very not sample efficient if we compare with other GP approaches that are not optimizing constants that way. I fear this may be the part of the optimization model that is making most of the job in terms of results.

As mentioned before, a big issue with the proposal is the lack of comparison with other semantic GP approaches, like the ones presented in Vanneschi et al. (2014), and more standard GP approach.

Vanneschi, L., Castelli, M., & Silva, S. (2014). A survey of semantic methods in genetic programming. Genetic Programming and Evolvable Machines, 15(2), 195-214.

There is a lack of evaluating the approach on benchmark problems for GP, like symbolic regression functions (see SRBench, https://github.com/cavalab/srbench). That would allow us to properly evaluate and compare performance with the state of the art.

The application contexts proposed are interesting uses of the approach, although I am not convinced they are of practical interest. Principled approaches are much preferable to learned ad hoc criterion, unless a clear added value is presented (this is not the case here). It is even stated at the end of sec. 6.1 that the learned optimizer is not generalizing outside the task suite, exposing some kind of overfitting. That limits greatly the interest for such an approach.


**Strength And Weaknesses:**

Strength
- Straightforward proposal for a novel semantic GP mutation operator.
- Apparently good empirical performances in three applicative contexts.

Weaknesses
- The proposal is relatively simple and incremental considering the work presented so far on semantic GP. There is no detailed overview of semantic GP, nor a good analysis of the similarity and differences of the current approach with the literature on the topic is missing.
- The experiments are done on specific problems and lack detailed comparison with other common semantic GP approaches.
- The paper writing quality is below average, and details on the method proposed are missing or unclear at times.
- The overall proposal appears relatively niched.


**Summary Of The Paper:**

The paper proposes a semantic mutation operator for genetic programming (GP), which aims at creating new diverse candidate solutions while keeping a good connection with the parent solutions. Semantic GP is a common approach that aims at exploiting the behavior of programs or subprograms (e.g., the output generated for given test cases used for evaluation) instead of their structure. This signal has been shown as relevant to be included in the optimization methods of GP, optimization guided purely on the structure of GP solutions being non-smooth or continuous, with simple changes in the structure leading to huge differences in the behavior. The proposed semantic mutation operator is presented and evaluated for three specific problems, that is real-valued function approximation, gradient-descent-based optimization function, and a loss function for reinforcement learning.

**Summary Of The Review:**

The proposal is simple, but the explanations are not very clear, many elements are not justified, there is a lack of comparison with other related methods and the problems tackled are non standard (not common benchmark in the field) nor of practical interest by themselves.

---

> ### Author Response · Authors · 2022-11-17
> **Response to reviewer zufj**
>
> Thank you for your detailed review, this is highly appreciated, and we have updated our paper with your suggestions.
>
> Regarding the weaknesses:
> * Our method is targeting genetic programming problems where the fitness evaluation is very expensive (e.g. neural network optimizers, RL agents), where evaluating the full fitness can take minutes or hours. Whereas the mutation objective in Equation (1) takes only a few seconds for the 25600 evaluations that we do. Our insight is that for expensive fitness functions, you can make genetic programming much more efficient by spending a relatively small fraction of compute on first optimising this much cheaper “fitness heuristic”.
> * On comparison with other semantic methods: we have empirical comparisons to similar methods in Figure 3. These include point mutation (Uy et al, 2009) and/or crossover (Nguyen et al, 2009). We also qualitatively compare our method and other methods in the Related Work section (see the paragraph “Semantic genetic programming”), and highlight how our mutation objective compares with previous methods (the explanation below objective (1) on page 3). It is possible that we have missed a method, and would happily add comparisons to other methods that are similar to ours if you can highlight it. (We surveyed Vanneschi et al. 2014 when writing this paper, and are not presently aware of any that are the most similar or likely to perform well that are not included.) For standard genetic programming methods, we found gplearn performed poorly, and did not include it in Figure 3 (see Section 4.2).
> * Regarding “The proposal is relatively simple and incremental” – (1) we see simple methods as a strength not a weakness (they are conceptually easier to understand, build upon, implement, and are more likely to generalise well), and (2) on incrementality, (a) we highlight that our method is 1-2 orders of magnitude more efficient in fitness evaluations compared with the previous closest methods (Figure 3); and (b) the diversity term we use is quite distinct from anything used previously; the most closest work is ensuring new programs are semantically not equal to previously evaluated programs (see related work); this new form of diversity-in-mutation is critical for the performance (Figure 1). See also our response to kHGQ.
> * On the proposal being niche: the ideas are applicable for any genetic programming setup where (1) the fitness evaluation is expensive, and (2) it is possible to define a semantic distance between programs. We give two example applications, but there are likely many more. The form of the mutation objective in Equation (1) is very generic, requiring only the definition of semantic distance between programs, and in fact is more generic than many other semantic methods (such as Nguyen et al 2009; Beadle & Johnson 2008; Uy et al 2009a;b), since we only require program-level semantic information, rather than node-level semantic information.
>
> On the smaller points:
> * The “G” in the definition of d(J, K) is the graph being mutated. We agree this is unclear and we’ve updated the description. Note that one occurrence of d in (1) is as d(M(G), H).
> * We’ve added more justification for using “min” rather than “sum” in the second term in Equation (1). In short, we want the terms to be balanced, whereas if we used a sum, then the mutation operator may give solutions which are far from most programs, but fail to satisfy the first term.
> * On using the same scaling \mu between the different terms: we also scale with \beta in the third term. In fact, although the first two terms look distinct and appear like they could have different scalings, they are two sides of the same coin: they require that the mutated program is at least distance \mu from all programs (including the original), but ideally not much further than \mu from the original. One should think of \mu as a learning rate.
> * We have given an expanded description of the raw mutation operator in the appendix. We found that the precise details of this did not make a statistically significant difference to the performance of the final mutation operator: to some degree, the mutation objective “smooths out” whatever set of raw programs it is given to optimise. Regarding the gating constant c, we mean that we pick the initial value of c according to the expression; for example, in z*c we take c=1, and in z+c we take c=0.
> * Regarding the scaling “s_i = max{10^-5,min(|u_i|, |u_i-1|, |u_i+1|)}” used in CMA-ES: we agree this description could be better. We have rewritten this paragraph.
> * Regarding “With probability 0.2, this is re-evaluation…” this is only done for applications with noisy fitness measurements, such as in RL. We have updated the paragraph.
> * Regarding the paragraph on "Collecting sample inputs for the semantics…”: sample program inputs are required to define the semantics (i.e. distance function) of the programs.

---

### Official Review · Reviewer_6oRb · 2022-11-02

**Confidence:** 3
**Correctness:** 3
**Technical Novelty And Significance:** 2
**Empirical Novelty And Significance:** 2
**Recommendation:** 3

**Clarity, Quality, Novelty And Reproducibility:**

* The reviewer feels that the technical difference between the proposed method and existing semantic genetic programming methods is not clear. The technical contribution should be made clear.
* The advantage of the proposed method on meta-learning tasks against existing methods is not clear, although the authors mentioned such experimental comparison is difficult due to the cost of reproducing existing methods.
* The authors did not provide the code and did not report the detailed experimental settings. The reviewer thinks that it is hard to reproduce the experimental results.


**Strength And Weaknesses:**

[Strength]
* A novel mutation operation taking into account the semantic similarity is introduced.
* The effectiveness of the proposed method is demonstrated for not only simple problems but also meta-learning tasks.

[Weaknesses]
* The reviewer is not an expert on this topic and is not confident whether the proposed semantic distance between programs defined in (2) is novel or not. The technical novelty of the distance function should be clarified.
* In the proposed method, it seems that minimizing the objective of (1) and distance calculation requires generated programs' execution, and such cost should not be ignored. The reviewer is not sure that the experimental comparison with existing methods was fair. It would be better to compare the performance under the same number of program executions or computational budgets.
* In the experiments on learning an optimizer and learning a reinforcement learning loss, although the use case of the proposed method is demonstrated, the advantage of the proposed method on such tasks compared to other genetic programming methods is unclear.
* The concept of the proposed method can be understandable. However, it seems difficult to re-implement it because the authors did not provide the code and detailed information on the implementation.

[Minor Comments]
What does `sin_big` mean in Figure 1-4?


**Summary Of The Paper:**

This paper presents a mutation operation to generate semantically close and diverse individuals in genetic programming. The proposed method generates a candidate program minimizing the mutation objective that takes into account the distance between parent, child, and previously evaluated programs. Then, the proposed method enables to the generation of a candidate program that is close but not too close to the parent and is not close to previously evaluated programs. To validate the effectiveness of the proposed concept, the authors apply the proposed method to simple regression, learning an optimizer, and learning reinforcement learning loss tasks. In the simple regression task, the proposed method can accelerate the search efficiency compared to existing mutation operations. In addition, the proposed method could find a better solution than existing algorithms in learning an optimizer and learning reinforcement learning loss tasks.

**Summary Of The Review:**

The topic treated in this paper is interesting and important. However, the reviewer feels that the technical novelty and empirical evidence of the advantage of the proposed method are unclear. Such weakness should be addressed to accept the paper.

---

> ### Author Response · Authors · 2022-11-17
> **Response to reviewer 6oRb**
>
> Thank you for your review and detailed comments.
>
> On novelty, please see our responses to the other two reviewers for a more detailed account of the ideas we build upon, and the ideas we introduce.
>
> Regarding the execution of the programs to calculate semantic distances: we target applications where fitness evaluation is expensive (for example taking minutes or hours), and in such applications the cost of executing a program can be orders of magnitude smaller (e.g. seconds for the multiple evaluations that we require to optimize Equation (1)). We go into more detail in our response to reviewer zufj.
>
> On experimental settings: we have updated some of the descriptions in the paper, for example, of the raw mutation operator. We are happy to update any other areas of the description of the algorithm in the paper that are ambiguous.

---

### Official Review · Reviewer_kHGQ · 2022-11-03

**Confidence:** 2
**Clarity, Quality, Novelty And Reproducibility:** The paper is written clearly with goo…
**Correctness:** 4
**Technical Novelty And Significance:** 2
**Empirical Novelty And Significance:** 2
**Recommendation:** 5

**Strength And Weaknesses:**

Strength:
- The paper provides several instantiations of the method, all seem quite interesting.

Weakness:
- The method seems pretty simple and not very surprising, so not sure how much conceptual novelty there is. I'm not an expert on genetic programming, so would leave it to other reviewers who have more domain expertise on this topic.
- For the learned optimizer experiment, there's no comparison with other methods. In fact there is a rich literature of previous works on this problem, i.e., how to select the hyperparameters in an algorithmic way. More comparison with them would be great.
- The experiments are definitely too small scale. E.g., for the learned optimizer, it's unclear whether it works on larger NN models, so the findings in this paper may be not relevant to modern deep learning.

**Summary Of The Paper:**

This paper proposes a new sampling method for genetic programming based on the semantics. The idea is that for each step, the algorithm would sample a new variant with different semantics (based on some distance function) that is also different from previous historical samples. Based on this idea, the authors demonstrate that this method improves sample complexity on a bunch of tasks, including function regression, learning an optimizer, and learning RL losses.

**Summary Of The Review:**

This paper mainly provide experiments on small scale datasets and lack comparison with existing methods. I'm not an expert on genetic programing, so I'm not sure how important these tasks are for the community --- from my perspective the tasks (e.g., learning optimizer and learning RL loss) are not super relevant these days especially when the experiments are only carried out on small datasets. Thus, I'll vote for rejection based on my current understanding, but would love to follow other reviewer's suggestion if they are more confident on this topic.

---

> ### Author Response · Authors · 2022-11-17
> **Response to reviewer kHGQ**
>
> Thank you for your review and comments.
>
> * On conceptual novelty (see also our response to zufj). We attempt a slightly more clear enumeration of the ideas we build upon, and the ideas we introduce:
>
>   1. *Existing idea:* Using semantic information. This is the idea of using the behaviour of (parts of) the program as an input to mutation, rather than the program syntactics (structure). See for example the survey of Vanneschi et al (2014).
>   2. *Existing idea:* mutations should be semantically close, but not too close, for efficient mutation. This is seen for example in Uy et al (2009a; b), Nguyen et al (2009), and Moraglio et al (2012).
>   3. *New idea:* using semantic information inside a continuous objective to optimise for (i.e. Equation (1)). Previous methods use semantic information in other ways, e.g., Uy et al. (2009a;b) do crossover by finding two semantically similar subtrees, which fails if no such pair exists (we can simply directly optimize for such a crossover, as in Section 5), or e.g. Moraglio et al who add together two programs and attempt to simplify (which suffers from program complexity blowup). Additionally, having a continuous objective allows us to trade off different constraints, such as semantic similarity versus program complexity.
>   4. *New idea:* using semantic information for program diversity *at mutation* (this is distinct from maintaining population diversity). When generating mutations, we avoid being similar to any previously evaluated program. The closest to this is to avoid re-evaluating programs that are semantically identical to a previously evaluated program (e.g. Alet et al., 2020; Real et al., 2020), but in continuous or large program spaces, the probability of two programs being identical is very small. This ew idea allows us to get very good performance, as shown in Figure 1.
>   5. *Newish idea:* Using semantic distance between *entire programs* rather than subnodes in a program. (At least, we are the first to apply this idea in an algorithm as far as we know.) This is more general (for example in some applications such as optimizers per-node semantic information is not meaningfully defined) and conceptually simpler (it is easier to think about what “distance between programs” means rather than “distance between nodes within a program”).
>
>   These individual ideas also hide the greater gestalt of what we provide: a conceptually simple framework (viz optimising an easy to understand mutation objective) that is widely applicable and highly sample efficient.
>
> * On experiments with learned optimizers: the intention here is to demonstrate the applicability of the method. We still discover novel features, such as weight expansion, and a new form of momentum, but agree that we have not yet demonstrated that the method can discover general-purpose SOTA optimizers. We hope to return to this in future work, but the main emphasis of the current paper is on introducing the method.

---

### Decision · Program_Chairs · 2023-01-20

**Decision:**

Reject

**Justification For Why Not Higher Score:**

Overall the reviewers find this work to be of interest to audience in the area of GP, and it is recognized that the authors showed good performance in several specific contexts. Nonetheless, the overall novelty and scope of this work is limited, and the authors are encouraged to take the comments from the discussion to further enhance this work for future submission.

**Justification For Why Not Lower Score:**

NA

**Metareview: Summary, Strengths And Weaknesses:**

This work proposed a novel semantic genetic programming mutation operator, and demonstrated its effectiveness using a set of applications in machine learning. Overall this work provide an interesting idea that would help to improve GP in several contexts, but the reviewers have overall reservations about this work in that: the proposed method is relatively straightforward, the comparisons and experiments could be significantly enhanced. It will be helpful for the authors to take these comments and improve their work.